# Nucleocapsid protein-specific monoclonal antibodies protect mice against Crimean-Congo hemorrhagic fever virus

Aura R. Garrison [1] ✉, Vanessa Moresco[2], Xiankun Zeng [1], Curtis R. Cline[1], Michael D. Ward[1], Keersten M. Ricks[1], Scott P. Olschner[1], Lisa H. Cazares[1], Elif Karaaslan [3], Collin J. Fitzpatrick[1], Éric Bergeron [3], Scott D. Pegan[2,4] & Joseph W. Golden [1] ✉

Crimean-Congo hemorrhagic fever virus (CCHFV) is a WHO priority pathogen. Antibody-based medical countermeasures offer an important strategy to mitigate severe disease caused by CCHFV. Most efforts have focused on targeting the viral glycoproteins. However, glycoproteins are poorly conserved among viral strains. The CCHFV nucleocapsid protein (NP) is highly conserved between CCHFV strains. Here, we investigate the protective efficacy of a CCHFV monoclonal antibody targeting the NP. We find that an anti-NP monoclonal antibody (mAb-9D5) protected female mice against lethal CCHFV infection or resulted in a significant delay in mean time-to-death in mice that succumbed to disease compared to isotype control animals. Antibody protection is independent of Fc-receptor functionality and complement activity. The antibody bound NP from several CCHFV strains and exhibited robust cross-protection against the heterologous CCHFV strain Afg09-2990. Our work demonstrates that the NP is a viable target for antibody-based therapeutics, providing another direction for developing immunotherapeutics against CCHFV.

Crimean–Congo hemorrhagic fever virus (CCHFV) is an enveloped virus and a member of the genus *Orthonairovirus* in the *Nairoviridae* family[1–4]. CCHFV infects a large number of wild and domesticated mammalian species, including giraffes, buffalo, zebra, bovines, and ovines, in addition to some avian species such as ostriches. However, infection in these animals is generally asymptomatic, at most producing a prolonged (>5 days) viremia[5,6]. In marked contrast, CCHFV infection in humans can lead to an acute and potentially life-threatening disease termed Crimean–Congo hemorrhagic fever (CCHF)[2,7,8]. CCHFV is naturally spread through bites of ixodid ticks, primarily those of the genus *Hyalomma*. In addition, human infections can result from occupational or ceremonial exposure to infected animals during the slaughter of livestock such as ostriches, cattle, and sheep[3,9]. Healthcare workers are also at an increased risk of exposure to nosocomial infections, particularly in situations where CCHFV is unsuspected, such as during the first recorded infection in Spain in 2017[7,8,10]. The mortality rate of CCHF ranges from 3 to 30% and is suspected to depend on multiple factors including viral strain, speed of diagnosis, and access to emergency health care[2]. There are currently no licensed vaccines or therapeutics to prevent or treat CCHFV, although ribavirin may provide some limited therapeutic benefit[11].

CCHFV has a tripartite, negative-sense RNA genome comprising small (S), medium (M) and large (L) segments. The S segment

[1]United States Army Medical Research Institute of Infectious Diseases, Fort Detrick, MD, USA. [2]Division of Biomedical Sciences, University of California Riverside, Riverside, CA, USA. [3]Viral Special Pathogens Branch, Division of High-Consequence Pathogens and Pathology, Centers for Disease Control and Prevention, Atlanta, GA, USA. [4]Department of Chemistry & Life Science, United States Military Academy, West Point, NY, USA. ✉e-mail: aura.r.garrison.civ@health.mil; joseph.w.golden.civ@health.mil

encodes the nucleocapsid protein (NP); the M-segment encodes the glycoprotein open-reading frame (ORF) that is cleaved into two structural glycoproteins ($G_N$ and $G_C$) and nonstructural proteins (including the mucin-like domain and GP38) and the L segment encodes the RNA-dependent RNA polymerase[12]. Recent work revealed that antibodies against specific glycoprotein targets protect against lethal CCHFV infection in murine models[13–17]. Specifically, our group, subsequently confirmed by others, discovered that the GP38 viral protein is an important antibody target against CCHFV[14,16,17]. A non-neutralizing monoclonal antibody (mAb) targeting this protein protects against lethal infection in a post-exposure environment and protects against different strains of CCHFV[14,17]. Complete protection appears to require complement, but not other Fc functions[14]. The majority of murine-neutralizing antibodies, which target $G_C$, have failed to provide postexposure protection in murine systems[14,18]. However, some human antibodies targeting $G_C$ provide protection in mice when treatment is initiated one day prior to virus challenge, but not when given one day post challenge. An engineered bi-specific antibody targeting different $G_C$ epitopes can overcome this barrier and provide postexposure protection[13]. These studies support the development of antibodies as effective countermeasures against CCHFV.

Vaccines targeting the M-segment glycoproteins can protect animals against severe disease[19–23]. In addition, the NP functions as a viable vaccine target, with several NP-based vaccine studies indicating that immune responses against NP confer protection[23–27]. Because NP is an internal viral protein and not generally exposed on the viral surface, it was presumed that vaccine-facilitated protection was predominately mediated by a T-cell response. However, the NP of CCHFV is abundant and highly immunogenic, and anti-NP antibodies are readily produced during infection in survivors[28,29]. The presence of these antibodies is widely used for serosurveys and diagnostics[30–34]. A more recent study using B-cell-deficient mice suggested that antibodies were the important correlate of protection of NP-targeting vaccines[27]. However, direct evidence of anti-NP antibodies protecting against CCHFV infection have not been reported. Here, we investigated if an anti-NP murine monoclonal antibody could protect mice from severe disease. Our results indicate that an antibody, mAb-9D5, targeting the NP provides significant protection in two CCHFV adult mouse models, interferon knockout mice (IFNAR$^{-/-}$) and in type I interferon antibody-blockaded mice (IS). These two IFN-I disrupted mouse systems are widely used for CCHFV studies[13,14,19,21–23,26,35–38]. We further investigated the protective efficacy of mAb-9D5 against heterologous challenge and examined the role of Fc-domains in protection. Our findings identify the NP as a viable target for antibody-mediated protection.

## Results

### A monoclonal antibody targeting CCHFV NP protects mice against infection

To determine if NP-targeting antibodies could provide protection against CCHFV, two groups of 10 IFNAR$^{-/-}$ mice were injected with an anti-NP monoclonal antibody (mAb-9D5) or an isotype control antibody via the intraperitoneal (IP) route on day −1 and day +3 relative to challenge. On day 0, all mice were infected with 100 plaque-forming units (PFU) of CCHFV strain IbAr 10200 by the subcutaneous (SC) route. The mice were monitored for signs of disease, and group weights were taken daily. All isotype control group mice succumbed to infection by day 5 (Fig. 1A). In contrast, mice treated with the mAb-9D5 antibody were significantly protected with 50% survival. In addition to the increase in percent surviving in the mAb-9D5-treated group (50%) versus the control group, there was also a delay in death in the treated group, with a median survival of 14 days compared to 5 days in the isotype control group. These findings indicated that NP-targeting antibodies can protect mice from CCHFV infection.

### MAb-9D5 limits early viral spread to the liver and spleen and delays lesion development

Hepatic injury is a salient feature in CCHFV-infected humans and mice[2,39,40]. To more critically assess the protective efficacy of mAb-9D5, we conducted a serial time course study in mice treated with isotype control antibody or mAb-9D5. Liver samples from five mice were collected from both groups on day 4 (isotype control and mAb-9D5-treated) and because mAb-9D5 treated mice survived, samples from this group were also collected on days 8 and 12. Generally, mAb-9D5 delayed liver injury in infected mice compared to isotype control-treated mice on day 4 (Fig. 1B and Supplementary Fig. S1). Histopathological lesions in livers of isotype control CCHFV-infected mice taken on day 4 included multifocal areas of lytic necrosis, characterized by shrunken hepatocytes with hypereosinophilic cytoplasm and pyknotic and/or karyorrhectic nuclei. The pyknotic/karyorrhectic nuclei were arranged either as individualized cells (i.e., "single-cell necrosis") in clusters, or as large aggregates of cellular debris. The necrosis often associated with areas of inflammation. All mAb-9D5-treated mice had a clinical score of 0 on day 4 dpi and one animal in the control group had a clinical score of 1 (slightly ruffled appearance), on day 8 dpi one mAb-9D5-treated mouse had a clinical score of 1 (slightly ruffled appearance) and the remaining mice scored 0. On day 4, livers from mAb-9D5 mice had markedly reduced liver injury compared to the isotype control antibody-treated mice. However, by day 8, hepatic damage in mAb-9D5 mice was mostly indistinguishable from isotype control mice on day 4. We also observed hepatocyte vacuolation located diffusely across regions (i.e., portal to central veins) and consistent with microvesicular (small vacuoles) in day 8 mAb-9D5 treated mice, and, to a lesser extent in a single isotype control liver from day 4. Vacuolation was absent in mice treated with mAb-9D5 on days 4 and 12. Viral RNA indicated by in situ hybridization (ISH) was present in all mice; however, staining was more intense in day 4 isotype controls and days 8 and 12 mAb-9D5-treated mice, compared to day 4 mAb-9D5-treated animals (Fig. 1B).

Spleens of isotype control-treated infected mice had a depletion of lymphocytes and areas of lymphocyte apoptosis/necrosis, inflammation of red pulp, histiocyte infiltration, and fibrin deposition (Supplementary Fig. S2). In contrast, splenic lesions were largely absent in day 4 infected, mAb-9D5 treated mice. However, by day 8 and day 12 splenic lesions in mAb-9D5 treated mice were equivalent to or higher than those of the isotype control animals on day 4. Among those more severe was histiocyte infiltration in the red pulp of day 8 mAb-9D5 treated animals. In the spleen, ISH signal was mostly absent, except for a single mAb-9D5 mouse on day 4, but similar to control mice, viral RNA was prevalent at later time points (Supplementary Fig. S2).

Our work has previously shown that mice infected with CCHFV and lacking IFN-I activity have nearly a complete loss of Kupffer cells, indicated by an absence of CLEC4F$^+$ staining on day 4 post infection[41]. Here, we also observed an extensive loss of CLEC4F$^+$ staining on day 4 in CCHFV-infected mice treated with the isotype control antibody, which was associated with intense staining for the viral NP (Fig. 1C). However, Kupffer cells were not lost on day 4 in mAb-9D5 treated mice, and NP staining was only minor. On day 8, Kupffer cells were mostly absent in mAb-9D5-treated mice, which coincided with high levels of NP. By day 12, the Kupffer cell population was restored, but viral antigen was still present, albeit in lower amounts compared to day 8. We also observed that mAb-9D5 treatment delayed the infiltration of CD68+ macrophages and CD45+ immune cells (Fig. 1D). Similar delays were seen with MPO+ granulocytes and Ki67+ proliferating cells (Fig. 1D).

### MAb-9D5 does not protect against postexposure

NP appears to be highly abundant during CCHFV infection in cell culture[42]. Therefore, we examined the day 4 serial time point isotype

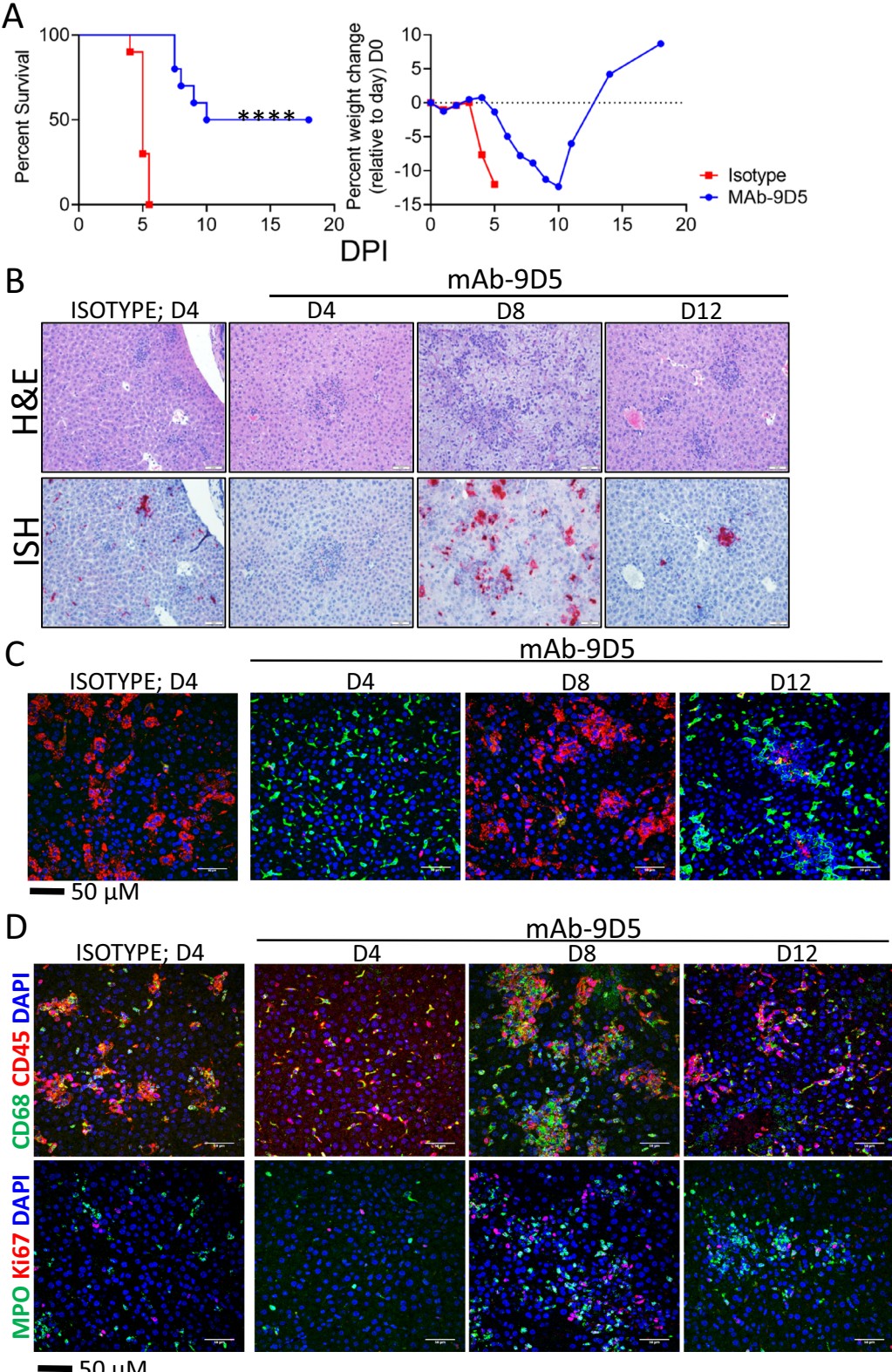

**Fig. 1 | MAb-9D5 protects mice from CCHFV infection. A** IFNAR$^{-/-}$ mice ($N$ = 10/group) were treated with mAb-9D5 (blue circles) or isotype control antibody (red squares) IP on days −1 and +3. Mice were challenged SC with CCHFV strain IbAr 10200 on day 0 and survival and group weight loss were monitored and plotted using Prism software. ****$P$ < 0.0001. Source data is provided. **B** Representative H&E ISH staining of livers of CCHFV strain IbAr 10200-infected mice treated with mAb-9D5 (days 4, 8, and 12) or isotype control antibody (day 4). ISH-stained tissue was counterstained with hematoxylin. **C** Liver sections from CCHFV strain IbAr 10200-infected mice treated with anti-NP or isotype control were stained with anti-CLEC4F (green) and anti-CCHFV NP antibodies (red). Cell nuclei were stained with DAPI (blue). **D** IFA straining for CD68+ macrophages (green) and CD45+ leukocytes (red) or Ki67+ proliferating cells (green) and MPO+ neutrophil granulocytes (green) in livers of anti-NP or isotype control-treated and -infected mice. Nuclei are stained with DAPI (blue). μM micrometer. **B**−**D** $N$ = 5 mice/group for days 4 and 8, and $N$ = 2 mice for day 12. Samples with the most severe pathology are shown.

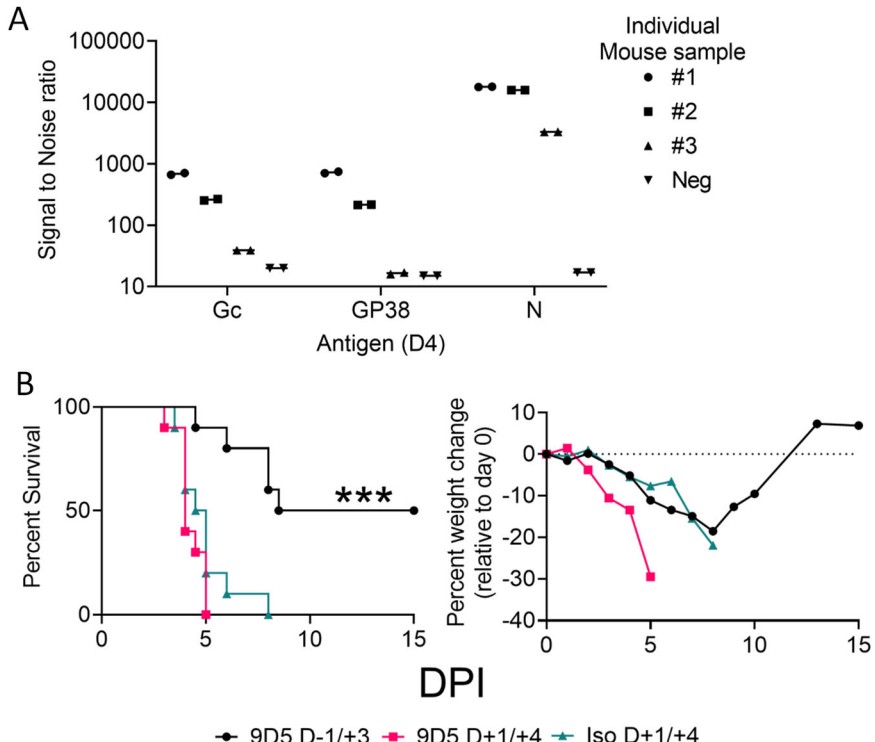

**Fig. 2 | MAb-9D5 does not provide postexposure protection. A** IFNAR[−/−] mice
(*N* = 3) (circles, squares and right side up triangles) were euthanized four days post
infection with strain IbAr 10200 or uninfected (*N* = 1) (upside down triangles), and
the presence of NP, $G_C$, and GP38 proteins in the sera was determined by MAGPIX
assay. The line shows the mean of the two displayed replicates. **B** IFNAR[−/−] mice
(*N* = 10 per group) were treated with two doses of the mAb-9D5 (1 mg/dose) on days

−1/ + 3 (black circle) or +1/ + 4 (red square) or isotype control antibody days +1/ + 3
(aqua triangle) and survival and group weight monitored. Mice were SC-infected
with CCHFV strain IbAr 10200 on day 0 and survival and weight monitored. Log-
rank test comparing each treatment group to the isotype control; ***P = 0.0004.
Source data is provided.

control animals (Fig. 1B) to determine the extent to which cell-free
NP was present in the circulation. For this, we used a sensitive
antigen capture system using MAGPIX to examine the presence of
NP in the cell-free serum (Fig. 2A). This study revealed that secreted
NP was readily detected in the serum of infected animals in addition
to glycoproteins Gc and GP38. Of these proteins, levels of NP
appeared to be the highest. Because NP was detected at high levels
in the serum of all three infected mice, we wanted to determine if
this may confound the post-challenge protective efficacy of the anti-
NP antibody. Mice (*N* = 10/group) were treated with two doses of
mAb-9D5 either on days −1/ + 3 or days +1/ + 4 relative to infection,
and a control group was treated on day +1/ + 4 with a nonspecific
antibody (isotype control) (Fig. 2B). Isotype control-treated mice
succumbed to infection by day 5 and exhibited precipitous weight
loss throughout infection (Fig. 2B). As above, mice treated with
mAb-9D5 on days −1/ + 3 exhibited a delayed weight loss compared
to control mice and 50% of the mice survived, which compared to
the isotype control group was significant (log rank; *P* = 0.0004).
Mice treated with mAb-9D5 post infection had less weight loss
compared to control mice but were poorly protected from lethality
and succumbed by day 8. These findings indicated that while mAb-
9D5 protects against CCHFV when given prophylactically, post-
exposure protection was limited.

**CCHFV NP was detected on the surface of infected cells in vitro**
To gain a mechanistic understanding of how NP-targeting antibodies
protect against CCHFV, we first demonstrated that mAb-9D5 does
not neutralize virus in a plaque reduction assay, in contrast to an
anti-$G_C$ antibody, a well-established neutralizing antibody target[43,44]
(Supplementary Table S1). We then investigated if the NP could
localize to the surface of CCHFV-infected cells. A549 cells were

infected with an MOI of 1 with CCHFV IbAr 10200 for 24 h and
stained for NP or $G_C$ under non-permeabilizing conditions and
compared to staining under permeabilizing conditions (Fig. 3 and
Supplementary Fig. S3). NP and $G_C$ were detected under both con-
ditions. The NP pattern in the non-permeabilized cells was distinct
from the permeabilized cells, the latter maintained the perinuclear
staining previously shown[45].

**mAb-9D5-mediated protection does not require Fc functionality**
Given that NP was detected on the surface of CCHFV-infected cells in
vitro, we next evaluated if Fc-mediated processes, such as antibody-
mediated cytotoxicity (ADCC) or complement-mediated functions,
were important for protection conferred by mAb-9D5. For this
experiment, we evaluated protection using Fc-receptor-deficient
(FcR[−/−]) and C3 deficient (C3[−/−]) mice or wild-type control mice
(Fig. 4). FcR[−/−] and C3[−/−] mice are unable to facilitate Fc-receptor
function or complement-mediated activity, respectively. Because
CCHFV only causes severe disease in mice deficient in IFN-I signaling,
infected mice were treated with an antibody to block IFN-I signaling as
previously described[21,41]. In this system, the kinetics of disease is
identical to IFNAR[−/−] mice. Mice (*N* = 10 per group) were challenged
with 100 PFU of CCHFV strain IbAr 10200 by the IP route. Mice were
treated with mAb-9D5 or an isotype control antibody on days −1/ + 3 by
the SC route. On day +1 post infection, mice were injected IP with mAb-
5A3 to disrupt IFN-I activity. The majority of FcR[−/−] mice were sig-
nificantly protected by mAb-9D5 (log rank; *P* < 0.0001), but not by
the isotype control antibody. Mice lacking C3[−/−] (which were on a
C57BL/B6:129 background) were also protected from CCHFV infection
to the same degree as wild-type control mice (log rank; *P* = 0.0201).
These data suggest that ADCC or complement are not major con-
tributors to protection.

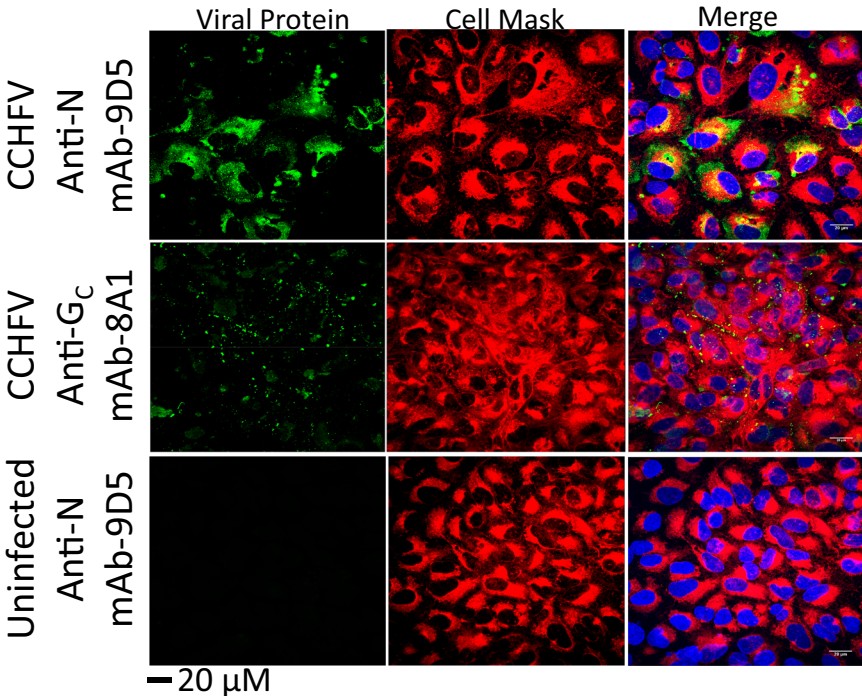

**Fig. 3 | Surface localization of NP in CCHFV-infected cells.** Non-permeabilized A549 cells infected with CCHFV strain IbAr 10200 were stained with the indicated antibodies against CCHFV viral proteins (NP; mAb-9D5 or Gc; mAb-8A1; green) and cell mask (red). Cell nuclei were stained with DAPI (blue). Representative images are shown from two experiments, each with six replicates. μM micrometer.

## Identification of the mAb-9D5 epitope and cross-reactivity against other CCHFV strains

We next used a well-established method to perform liquid chromatography-mass spectrometry (LC-MS) and protease protection studies to identify the region of NP bound by mAb-9D5[46–49]. MAb-9D5 was bound to magnetic beads, and the recombinant NP was bound to the antibody, cross-linked, and then digested with trypsin/Lyc C. As a control, a non-cross-linked sample was also analyzed. This analysis identified the mAb-9D5-protected region between amino acids 184 and 208, suggesting these amino acids encompass the antibody epitope (Fig. 5 and Supplementary Fig. S4). This region contains a concentration of several charged residues comprised within helix α8, which is located in the stalk portion of the NP and a flexible linker region that connects to the head region (Fig. 5A). NP, encoded on the S segment, exhibits high homology between CCHFV strains in comparison to the proteins encoded on the M and L segments[50]. The mAb-9D5 epitope identified in this study is highly homologous between several diverse strains of CCHFV isolated throughout the world (Fig. 5B and Supplementary Table S2). MAb-9D5 was produced against strain IbAr 10200, a laboratory-adapted strain isolated from a tick feeding on a camel[43], and our mouse studies above examined protection against this homologous strain. To evaluate the cross-protective potential of mAb-9D5, BioLayer Interferometry (BLI) was used to determine the binding affinity of mAb-9D5 for NP originated from a broad range of CCHFV strains. Afg09-2990 (Afg09) and Kosova Hoti (Clades IV and V) presented the highest nanomolar binding affinities (KD values), followed by Semunya (Clade II) (Fig. 5C). This is consistent with the fully conserved nature of the mAb-9D5 identified epitope between these three CCHFV strains (Fig. 5B). NP from IbAr 10200 and Senegal strains (Clades III and I) showed lower KD values. The protein originating from the closely related Aigai virus possessed a KD with mAb-9D5 similar to that observed from the Semunya strain.

To reveal the impact of strain-strain differences of other identified NP antibodies compared to mAb-9D5, several anti-NP mouse ascites-mAbs were screened against a similar array of NP using BLI (Supplementary Fig. S5 and Supplementary Table S3). We used a recently established methodology[51] that allows for rapid semi-quantitative screening of non-purified mAbs from animal fluid to assess antibody-antigen interactions. This methodology focuses on the measurement of wavelength changes (nm) during the association step and allowed insights into binding similarities within this broad range of mAbs and NPs. Binding magnitudes for NPs from CCHFV stains, or the Aigai virus towards mAb-9D5, mAb-9A1, and mAb-5F4 were consistent. mAb-7E8 and mAb-2G9 had marked differences in binding between the same NPs. While mAb-7E8 bound to all CCHFV strains, no measurable binding event occurred with the Aigai NP. Also, mAb-2G9 only showed a relatively strong binding event with the NP from the Senegal strain of CCHFV. As expected, due to its evolutionary distance from CCHFV[52], Erve virus NP did not bind to any of the tested mAbs and functions as a negative control (Supplementary Fig. S5 and Supplementary Table S3). These results indicated that NP-targeting CCHFV mAbs have broad spectrum cross-binding ability.

## MAb-9D5 protects against heterologous strains of CCHFV

Because mAb-9D5 bound Afg09 with the highest affinity, we evaluated its ability to protect mice against this strain. Both strains IbAr 10200 and Afg09 are lethal in IFN-I deficient mice, and kinetics of infection are identical[14,22]. Ten mice per group were treated IP with 1 mg of mAb-9D5, or isotype control, on day −1 and day +3 relative to SC challenge with 100 PFU of CCHFV strain IbAr 10200 or Afg09. The antibody treatment against IbAr 10200 challenge resulted in 40% survival and 80% survival against Afg09, compared to the isotype controls for each virus strain (Fig. 6). Both mAb-9D5 treatment groups had similar weight loss curves, which was delayed compared to the isotype control groups (Fig. 6). The protective efficacy of the antibody treatment was significant against both strains. These findings indicated that mAb-9D5 has the ability to protect against heterologous CCHFV strains, and protection may be enhanced due to higher binding affinity to Afg09 NP.

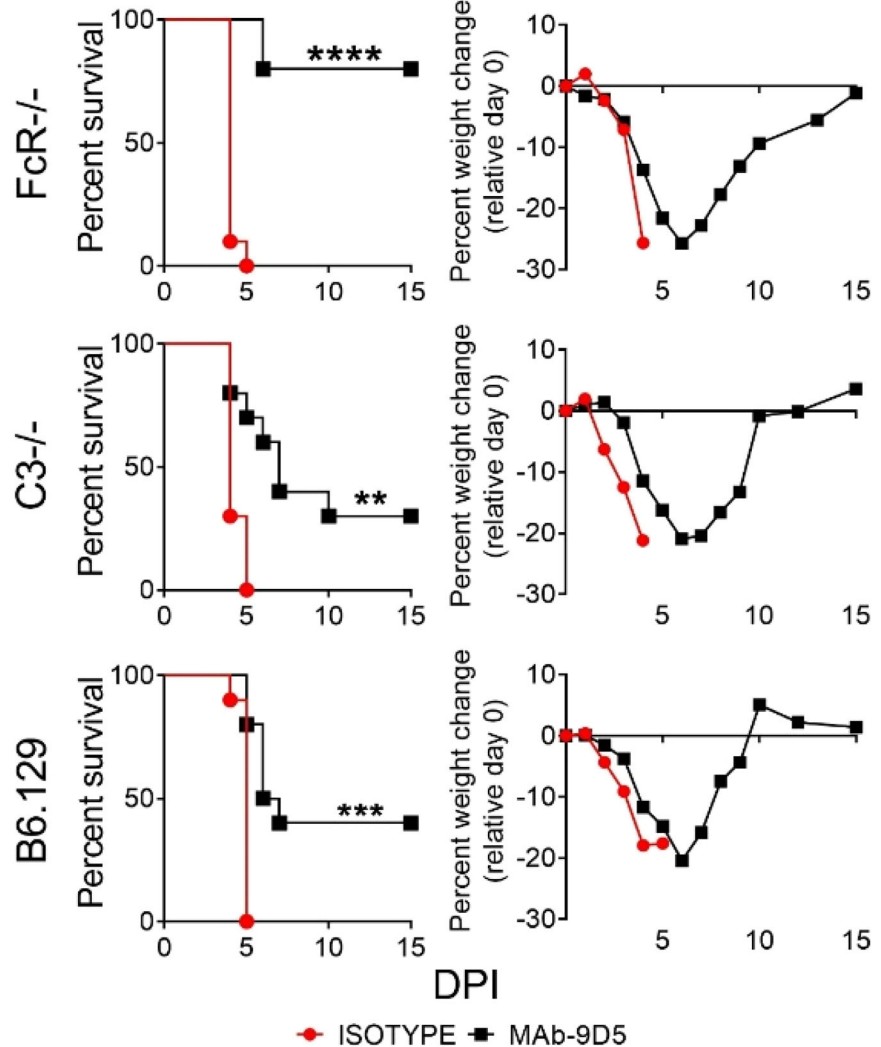

**Fig. 4 | Fc-domains do not impact the protective efficacy of mAb-9D5 in IFN-I antibody-blockaded mice.** FcR$^{-/-}$, C3$^{-/-}$, or B6:129 (wild-type) mice ($N$ = 10/group) were injected SC with two doses of the mAb-9D5 (black squares) or an isotype control antibody (red circle) (1 mg/dose) on D-1/ + 3. IFN-I was blocked on day +1 using mAb-5A3 (2.5 mg) injected IP. Mice were challenged with CCHFV strain IbAr 10200 by the IP route. Survival and group weights were plotted. Log-rank test; ****$P$ < 0.0001, ***$P$ = 0.0004, **$P$ = 0.0012. Source data is provided.

## Discussion

Because NP is a highly abundant internal protein found in virally infected cells and believed to be buried within viral particles it is often overlooked as a target of therapeutic antibodies against viruses compared to glycoproteins, which are well-known to be exposed on viral envelop and cell surfaces[53]. Nevertheless, NP is becoming more appreciated as an important target for antibody-based treatments. NP-targeting antibodies have been shown to protect against several virus groups, including influenza, arenaviruses, and coronaviruses[54–58]. Here, we identify CCHFV NP as an important protective antibody target for CCHFV infection. Until this study, NP-targeting mAb-9D5 was used as a laboratory reagent for CCHFV immunofluorescent staining and ELISAs[13,59–61]. CCHFV NP-targeting vaccines have shown promise and our findings here also add to the evidence that humoral immunity may be a critical component in this protection[27]. NP is produced in extremely high amounts during infection and is the most prominent target of the humoral response in CCHFV-infected individuals. Our data show that NP is present in the cell-free serum of infected mice. We believe this may have hampered the ability of mAb-9D5 to protect animals post infection, as the antibody was likely quickly inactivated by soluble NP. Healthcare workers are extremely vulnerable to CCHFV exposure from infected patients making an urgent need for anti-CCHFV products that can prevent nosocomial spread[7,8,10]. Thus, while mAb-9D5 did not exhibit robust protection when given postexposure, it may have important utility as a prophylaxis. While no other anti-NP antibodies were tested here, our findings warrant further screening to identify molecules that may confer postexposure protection. To this end, we identified other anti-NP antibodies that cross-react with different CCHFV NP (Supplementary Fig. S5). It is possible these other antibodies will yield higher postexposure efficacy.

The mechanisms by which NP antibodies disrupt CCHFV are unclear. We observed NP on the surface of infected cells, yet our findings failed to show convincing evidence that complement or Fc-mediated function (ADCC) were responsible for protection. This is similar to work with lymphocytic choriomeningitis virus (LCMV), an arenavirus, where both Fc-receptors and complement are dispensable for protection[58]. Curiously, LCMV NP can be found on infected cell surfaces and on virions[62,63]. We are currently examining CCHFV envelope for the presence of NP. There are important differences between LCMV and CCHFV anti-NP protection. LCMV-infected mice can be protected postexposure by anti-NP antibodies. More recent work suggests that anti-LCMV NP antibodies facilitate protection by enhancing NP interaction with TRIM21, a cytosolic antibody binding receptor and E3 ubiquitin ligase. TRIM21 binds antibody Fc-regions

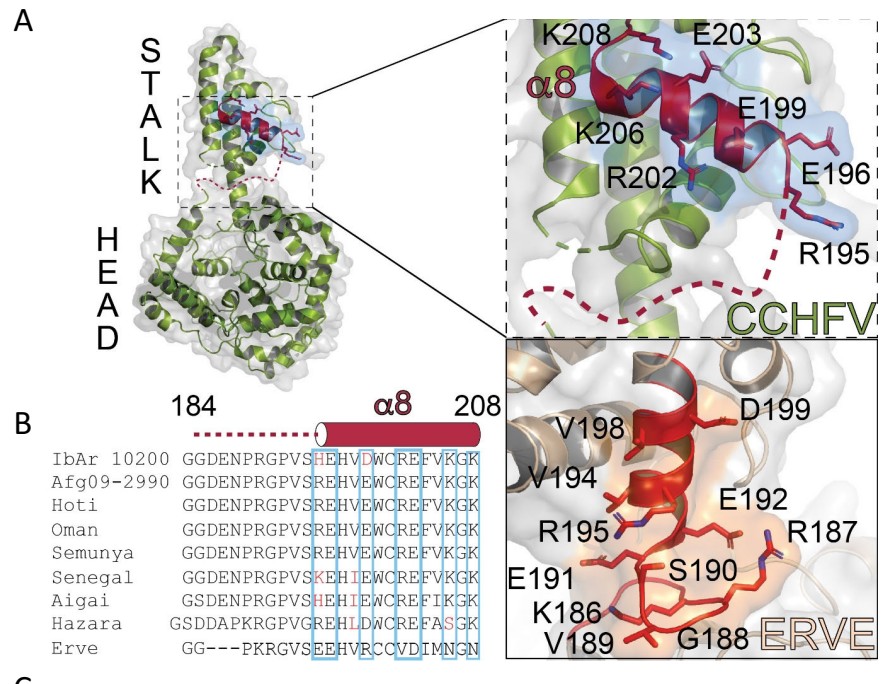

**Fig. 5 | Binding domain of mAb-9D5 on the NP. A** Structural representation of the NP monomer. The dashed box indicates the magnified NP region depicting the mAb-9D5 binding region identified by LC-MS. **B** Epitope amino acid comparisons between diverse strains of CCHFV and related nairoviruses IbAr 10200 (accession #MH483987.1), Afg09 (accession #HM452305.1), Kosova Hoti (accession #JN173797.1), Oman (accession #DQ211645.1), Semunya (accession #DQ076413), Senegal (accession #DQ211640), Aigai virus (accession #NC_078226), Hazara virus (accession #NC_038711) and Erve virus (accession #JF911699). **C** BLI binding kinetics of mAb-9D5 hybridoma with nairovirus NPs. BLI SA biosensors were loaded with NP (antigen) at concentration of 500 nM. Anti-NP mAb was evaluated in triplicate at concentrations of 700, 350, and 100 nM. NB not binding. Source data is provided.

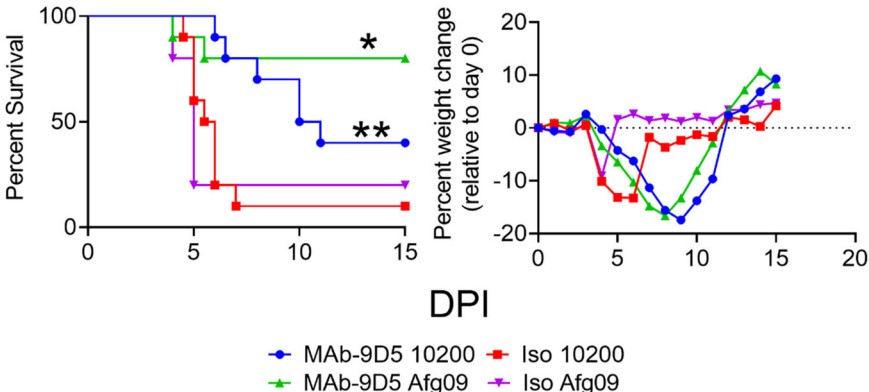

**Fig. 6 | Heterologous protection of mice by mAb-9D5.** IFNAR$^{-/-}$ ($N = 10$) mice were treated with two doses of the mAb-9D5 (blue circles; green triangles) or an isotype control antibody (red squares; purple triangles) on day −1/ + 3, and challenged SC with either IbAr 10200 (blue circles, red squares) or Afg09 (green triangles, purple triangles). Survival and percent group weight change were monitored. Log-rank test; **$P = 0.0055$, *$P = 0.011$. Source data is provided.

within the cytoplasm and directs antigen–antibody complex to proteasomal degradation[64,65]. Degraded NP is then displayed to the immune system, where it hastens cytotoxic T-cell responses[54]. It is unlikely this process is functioning in the CCHFV system. If it were, we would expect to have observed protection when the antibody was given 1 day post challenge. Furthermore, the infection phase in the LCMV murine system is longer compared to CCHFV, giving time for the CTL T-cell response to control disease. TRIM21 can also directly inactivate nonenveloped viruses entering the cytoplasm with bound antibody[66–68]. More studies are needed to resolve the mechanism, including any role for TRIM21, by which anti-NP CCHFV antibodies protect prior to exposure, but not after infection.

Structurally, CCHFV NP is comprised of a globular head domain and a flexible arm. The mAb-9D5 epitope was identified by mass spectrometry and is located within a region of the NP containing high structural flexibility[69]. This region is highly conserved and consistently we found that mAb-9D5 bound NP from all the CCHFV strains tested, as well as Aigai virus, a different nairoviruses species closely related to CCHFV[70]. Interestingly, mAb-9D5 bound Afg09 and Hoti NP with the highest affinity compared the strain IbAr 10200, the strain used to produce the antibodies. Indeed, mAb-9D5 protected against Afg09 better than against IbAr 10200. Considering the few amino acid differences found within the mAb-9D5 epitope of CCHFV strains and that of the Aigai virus, this implicates variations at solvent-pointing positions 195 and 199 as likely sources of the variations in affinity observed. Specifically, longer or higher charged residues such as those in CCHFV strains Afg09, Hoti, and Semunya are likely more favorable to mAb-9D5 binding (Fig. 5C). Not surprisingly, with the low homology within the epitope region of the Erve virus within this region, Erve NP had no binding affinity against mAb-9D5. Nevertheless, these binding and heterologous protection data predict that anti-NP immunotherapeutics will likely cross-protect broadly against CCHFV and Aigai virus, but not more broadly against more distantly related nairoviruses.

Immunotherapeutics represent an important class of medical countermeasures against viruses, including poliovirus[71], orthopoxviruses[72], rabies virus[73], Ebola virus[74], Junin virus[75], and SARS-CoV-2[76]. We previously demonstrated that anti-GP38 antibodies can protect mice postexposure[77]. Other studies have shown that bi-specific antibodies targeting $G_C$ can also protect against infection[13]. It is possible that antibody combinations that target NP and GP38 (or $G_C$) would provide synergistic protection and function as both potent prophylaxis and postexposure therapeutic. There is a growing concern that members of the *Bunyavirales* order have pandemic potential. Indeed, CCHFV is endemic on three continents, Africa, Asia, and Europe. It is also an important emerging pathogen due to the continued geographical spread of its vector, the *Hyalomma* ssp ticks, with a recent human fatal case reported in Spain[7]. Future work should begin focusing on deploying these immunotherapeutics to help mitigate severe CCHF and quash any nosocomial or close contact infections. Coupled with rapid diagnostics[78], these antibody-based drugs may represent the most effective method to control CCHFV and limit its pandemic potential.

## Methods

### Ethics statement
All animal studies were conducted in compliance with the Animal Welfare Act and other federal statutes and regulations relating to animals and experiments involving animals and adheres to principles stated in the Guide for the Care and Use of Laboratory Animals, National Research Council[79]. All animal experimental protocols were approved the United States Army Medical Research Institute of Infectious Disease Research (USAMRIID) Institutional Animal Care and Use Committee (IACUC). The facilities where this research was conducted are fully accredited by the Association for Assessment and Accreditation of Laboratory Animal Care International.

### Mice
C57BL/6 (BL6), IFNAR KO mice (B6.129S2-Ifnar1tm1Agt/Mmjax), Bl6;129 mice, and C3 knockout mice (B6;129S4-C3tm1Crr/J) were purchased from The Jackson Laboratory. Fc-receptor KO mice (C.129P2(B6)-Fcer1g$^{tm1Rav}$ N12) were obtained from Taconic. Mice were all female and 7–9 weeks in age at the time of challenge. Mice were maintained in an environment with 12 h light/dark cycle, 68–79 °F (set point: 74.5 °F), and 30–70% humidity.

### Virus and cells
HepG2 cells (ATCC; HB-8065) were propagated in Modified Eagle's Medium with Earl's salts (MEM)(Corning) and Huh7 (Texas Biomedical Research Institute), and SW13 (ATCC; #CCL−105) cells were propagated in Dulbecco's Modified Eagles Medium with Earle's Salts (DMEM) (Corning). Media were supplemented with 10% fetal bovine serum (FBS) (Gibco), 1% penicillin/streptomycin (Gibco), 1% sodium pyruvate (Sigma), 1% ʟ-glutamine (HyClone), and 1% HEPES (Gibco). A549 cells (ATCC; #CCL-185) were propagated in Hams F-12 media supplemented with 10% FBS, 1% glutamax, 1% penicillin/streptomycin, and 1% nonessential amino acids. Minimally passaged CCHFV strain Afg09-2990 (Afg09)[80] or strain IbAr 10200 (USAMRIID collection) were used for all experiments as indicated. Afg09-2990 was previously passaged three times in Vero cells and propagated two times in Huh7 cells, and IbAr 10200 was previously passaged nine times in suckling mouse brain and then propagated three times in HepG2 cells. The viruses were collected from clarified cell culture supernatants, sequenced to confirm identity and free of contamination, and stored at −80 °C. All CCHFV work was handled in BSL-4 containment at USAMRIID.

### Anti-CCHFV and isotype antibodies
Anti-CCHFV murine mAbs are part of the USAMRIID hybridoma collection produced by Jonathan F. Smith. As previously described, the USAMRIID anti-CCHFV murine mAb were developed by immunizing BALB/c mice with CCHFV-infected suckling mouse brain homogenates, and the B-cell hybridomas were generated by the fusion of Sp2/0 mouse myeloma cells with the splenocytes from the immunized mice[43]. The mAb-9D5 is an isotype IgG2a. Antibodies for murine challenges were purified in-house using the USAMRIID hybridoma facility. Murine isotype control antibodies were also provided by the USAMRIID hybridoma facility.

### Passive protection experiments
Mice were challenged with 100 PFU of CCHFV strain IbAr 10200 or Afg09 by the subcutaneous (SC) (IFNAR$^{-/-}$) or intraperitoneal (IP) (all other mice) route as indicated. Virus was diluted in a total volume of 0.2 ml PBS. All mice except IFNAR$^{-/-}$ were IP injected with 2.5 mg of anti-IFNR1 (mAb-5A3) (Leinco Technologies, Inc; I-401) diluted in PBS 24 h post infection in a total volume of 0.4 ml. For mAb-9D5 antibody injections, mice were injected SC or IP with 1 mg/antibody/dose in a total volume of 0.2 ml diluted in PBS, as indicated.

### Histology
Necropsy was performed on the liver and spleen. Tissues were immersed in 10% neutral buffered formalin for 30 days. Tissues were then trimmed and processed according to standard protocols[81]. Histology sections were cut at 5–6 μM on a rotary microtome, mounted onto glass slides and stained with hematoxylin and eosin (H&E). Examination of the tissue was performed by a board-certified veterinary pathologist.

### In situ hybridization
CCHFV was detected in infected liver samples by ISH probes targeting IbAr 10200 or Afg09 M-segment of CCHFV as previously reported[41]. Formalin-fixed paraffin-embedded (FFPE) liver sections were deparaffinized and peroxidase blocked. Sections were then incubated with

ISH probes at 40 °C for 2 h, rinsed, and the signal amplified by applying Pre-amplifier and Amplifier conjugated with HRP. A red substrate-chromogen solution was applied for 10 m at ambient temperature. The slides were further stained with hematoxylin. Images were captured on a Zeiss LSM 880 confocal system and processed using ImageJ software.

### IFA of tissues

FFPE tissue sections were deparaffinized using xylene and a series of ethanol washes. The sections were heated in Tris-EDTA buffer (10 mM Tris Base, 1 mM EDTA Solution, 0.05% Tween-20, pH 9.0) for 15 min to reverse formaldehyde crosslinks. After rinses with PBS (pH 7.4), the section was blocked with PBT (PBS + 0.1% Tween-20) containing 5% normal goat serum overnight at 4 °C. Then the sections were incubated with primary antibodies: rabbit polyclonal anti-myeloperoxidase (MPO) at a dilution of 1:200 (A039829-2, Dako Agilent Pathology Solutions), rat monoclonal anti-CD45 antibody at a dilution to 1:100 (05-1416, Millipore Sigma), rabbit polyclonal anti-CD68 at a dilution of 1:200 (ab125212, Abcam), and mouse monoclonal anti-Ki67 at a dilution of 1:200 (clone B56, BD Biosciences) for 2 h at room temperature. For CLEC4F and CCHFV NP detection, samples were incubated with a polyclonal goat anti-CLEC4F antibody at 1:20 dilution (PA5-47396; Thermo Fisher Scientific) and the anti-CCHFV NP murine monoclonal antibody MAb-9D5 protein at 1:500 dilution overnight at 4 °C. After rinses with PBT, the sections were incubated with secondary goat anti-rabbit or anti-chicken Alexa Fluor 488 at dilution of 1:500 (Thermo Fisher Scientific) and goat anti-mouse or anti-rat Alexa Fluor 568 at a dilution of 1:500 (Thermo Fisher Scientific) antibodies, for 1 h at room temperature. Sections were coverslipped using the Prolong Diamond mounting medium with DAPI (ThermFisher). Images were captured on a Zeiss LSM 880 or LSM700 (CLEC4F staining) confocal system (Zeiss) and processed using ImageJ software (National Institutes of Health).

### MAGPIX assay

**Capture bead preparation.** CCHFV monoclonal antibodies (mAbs) were covalently linked to magnetic microspheres following the manufacturer's instructions (Luminex) to capture NP (12E10), $G_C$ (11E7), and GP38 (13G8). Antibodies were from the USAMRIID monoclonal antibody repository. Briefly, 12.5 million microspheres were washed three times with 500 µL of activation buffer and resuspended in 274.5 µL of activation buffer. Next, 144.0 µL of sulfo-N-hydroxysulfosuccinimide and 81.5 µL of 1-ethyl-3-(3-dimethylaminopropyl) carbodiimide hydrochloride solutions were added and tubes were gently rotated for 20 min. After activation, microspheres were washed three times with coupling buffer, and antibody was added at 4 µg per million microspheres. The reaction was allowed to incubate for 2 h, after which the microspheres were washed three times with 500 µL of PBS-T (phosphate buffered saline with 0.05% Tween-20), resuspended at 12.5 million microspheres per mL in PBS-T, and stored at 4 °C. Each mAb was coupled to a spectrally distinct microsphere for ease of multiplexing.

**Detector antibody labeling.** CCHFV mAbs to detect NP (5G2), $G_C$ (8A8), and GP38 (9C6) were covalently labeled with biotin according to the manufacturer's instructions. Biotinylation of mAbs was achieved with the EZ-link™ Sulfo-NHS-LC-Biotin, No-Weigh™ Format kit (Pierce). Briefly, 50–60 µg of mAbs were reacted with a 20-fold molar excess of freshly prepared sulfo-NHS-LC-biotin (sulfosuccinimidyl-6-[biotinamido]hexanoate) at room temperature for 30 m. After the reaction, excess biotin was removed by dialyzing against PBS-T.

**General assay procedure.** Assays were developed on the Magpix® platform using white, Costar, round-bottom 96-well plates. Plates were loaded with 2500 microspheres per well of each capture bead to create a triplex, placed on a magnetic block for 1 m, and manually decanted. Samples were diluted 1:20 in 5% skim milk in PBS-T (SM), and 50 µL was applied to each well in triplicate. Plates were then covered and allowed

to incubate with 450 rpm shaking for 1 h. After incubation, microspheres in each well were washed three times with 100 µL of PBS-T. Biotinylated detector mAbs were diluted to 4 µg/mL in SM and 50 µL was added to the appropriate wells. Plates were then covered and allowed to incubate with 450 rpm shaking for an additional hour. After incubation, microspheres were washed three times with PBS-T. For the fluorescent reporter, streptavidin phycoerythrin (SAPE; Thermo Fisher) was diluted to 10 µg/mL in SM and 50 µL was added per well before covering and incubating with shaking at 450 rpm for 30 m. After the final incubation, microspheres were washed three times with PBS-T, suspended in 100 µL of PBS-T, and read by the Magpix® instrument. Two replicates were run per sample.

### LC-MS epitope mapping

**Sample preparation.** Antigen–antibody complex was generated by adding 12 µg anti-NP mAb-9D5 to 15 µg CCHF NP and incubated for 1.5 h at room temperature. Pierce Protein A/G magnetic beads™ (25 µL) were prepared per manufacturer's instructions and the antibody/antigen mixture added followed by incubation at room temperature for 2 h. The beads were washed 3× with 1 ml 1x PBS followed by two washes with 500 µL 25 mM NaHCO$_3$, pH 8.0. Beads were then resuspended in 25 µL 25 mM NaHCO$_3$ and 25 µL of a 100 mM solution of (bis[sulfosuccinimidyl] suberate (BS$_3$) crosslinking reagent (Thermo Fisher 21580) added for a final BS3 concentration of 50 mM. Control reactions were performed with 25 mM NaHCO$_3$ only. The reaction was incubated on ice for 2 h and subsequently quenched for 15 m with 50 mM Tris-HCl pH 8.0. Beads were washed 3× with 500 µL 25 mM NaHCO$_3$, reduced with 10 mM DTT and alkylated with 15 mM iodoacetamide, and finally resuspended in 25 mM NH$_4$HCO$_3$ pH 8.0, 5% acetonitrile, 12.5 nM Trypsin/LysC (Promega V5071), and 1% ProteaseMAX™ trypsin enhancer. Digests were incubated overnight at 37 °C with mild shaking. Recovered digests were purified by C18 spin column, dried to completion and stored at −20 °C until ready for MS analysis.

**LC-MS/MS analysis.** Sample digests were resuspended in 30 µL of 0.1% formic acid. A Dionex 3000 RSLCnano system (Thermo Scientific) injected 5 µL of each digest onto a pre-column (C18 PepMap 100, 5 µm particle size, 5 mm length × 0.3 mm internal diameter) using a flow rate of 10 µL/m. The loading solvent was 0.1% formic acid in HPLC-grade water. Peptides were then loaded onto an Easy-Spray analytical column (15 cm × 75 µm) packed with PepMap C18, 3-µm particle size, 100 A porosity particles (Thermo.). A 2%–38% B gradient elution over 80 min was formed using Pump-A (0.1% formic acid) and Pump-B (85% acetonitrile in 0.1% formic acid) at a flow rate of 300 nL/min. The column eluent was connected to an Easy-Spray source (Thermo) with an electrospray ionization voltage of 2.2 kV. An Orbitrap Elite mass spectrometer (Thermo) with an ion transfer tube temperature of 300 °C and an S-lens setting of 55% was used to focus the peptides. A top 15 data-dependent MS/MS method was used to select the 15 most abundant ions in a 400–1600 amu survey scan (120,000 resolution FWHM at $m/z$ 400) with a full AGC target value of 1e6 ions and a maximum injection time of 200 ms higher-energy collisional dissociation (HCD) MS/MS spectra were acquired at a resolution of 30,000 (FWHM at $m/z$ 400) with an AGC target value of $5 \times 10^5$ ions and a maximum injection time of 200 ms. The isolation width for MS/MS HCD fragmentation was set to 2 Daltons. The normalized HCD Collision energy was 40% with an activation time of 0.1 ms. The dynamic exclusion duration was 30 s.

**Database search and protein identification.** Acquired MS/MS protein searches were performed with ProteomeDiscoverer 2.4 (Thermo) or PEAKS 7.4 (Bioinformatics Solutions) using a Orthonairoviridea (taxID 1980517 – 9 sequences) and human (taxID 9606–20,387 sequences) subset of the SwissProt_2017_01_18 database. Variable modifications used were carbamyl (KMR), methyl (DE), acetyl (K), deamidated (NQ),

and oxidation (M). Cysteine carbamidomethylation was specified as a constant modification. The peptide-level false discovery rate (FDR) was set at 0.1% using posterior error probability validation. Only peptides having an XCorr of >2.0 and at least two peptide spectral matches (PSM's) were considered, with both unique and razor peptides used for identification. Mass tolerances were 10 ppm for the MS1 scan and 0.6 Da for all MS/MS scans. For this experiment, two biological replicates were analyzed in triplicate. The LC-MS/MS data generated in this study have been deposited in the MassIVE database database under accession code MSV000092981 (https://massive.ucsd.edu/ProteoSAFe/static/massive.jsp).

### BLI assays mAbs binding to NP

**NP expression and purification.** Plasmid constructs expressing full-length NP from five CCHFV clades (Afg09, Hoti, IbAr 10200, Senegal, and Semunya), and the related viruses (Aigai-Pentalofos and Erve) were cloned into pET-28a (+) vector (GenScript). NP coding sequences were added downstream to the 8X His and glutathione S-transferase (GST) tags sequences and HRV3C protease cleavage site to allow the release of the NP from the fused tags. The plasmid inserts were codon-optimized for expression in BL21 (DE3) *Escherichia coli* competent cells (New England BioLabs). Protein expression and purification consisted of several steps, including immobilized metal affinity (IMAC) and size-exclusion (Superdex 200−Cytiva) chromatography and were performed as previously described[69].

**BLI kinetics assay mAb-9D5 to NP.** BLI assay was performed using GatorPrime (GatorBio) instrument at 30 °C with shaking at 1000 rpm. Streptavidin biosensors (Flex SA) were pre-hydrated in priming buffer assay (GatorBio) for 600 s and dipped into baseline assay buffer (K buffer) for 120 s. SA biosensors were then loaded with biotinylated NP (for all strains tested) at a concentration of 500 nM for 500 s, followed by a second baseline step (K buffer) for 240 s. For the association step, the loaded biosensors were dipped into purified anti-NP mAb-9D5 (hybridoma) at concentrations of 700, 350, and 100 nM in triplicates for 1000 s, followed by a dissociation step (K buffer) for 4000 s. mAb-9D5 binding kinetic values (KD) were obtained by fitting the association and dissociation curves of all concentrations tested (global fit) after subtracting controls (unloaded biosensors) using Gator Analysis software (v 2.7.3 1013).

**Mouse ascites-mAbs binding to different NPs.** A total of five mouse ascites-mAb (9D5, 9A1, 7E8, 5F4, and 2G9) binding to different NP strains were analyzed by BLI employing 1:64 dilutions (within the mAb-9D5 hybridoma range of concentration) in K buffer. The assay conditions were the same as mentioned above except for the second baseline step which consisted of 2% BSA in K buffer, as a blocking step for unspecific binding. Ascites-mAb binding to NP-loaded biosensors were analyzed in triplicate, including a reference control of unloaded biosensor (no NP). The values for ascites-mAb NP binding are expressed as wavelength shift values (nm) which are directly proportional to the amount of binding within a range of linearity. The average nm values at the endpoint association step for each NP were subtracted from the values at the beginning (2 and 4 s) after reference control subtraction and association curve fitting[51].

### Plaque reduction neutralization test

A dilution series from 1:20 to 1:640 of mAb-9D5, or the positive control mAb−11E7, or the negative control mAb-6D8 (anti-Ebola GP antibody) were incubated with either CCHFV strain Afg09 or IbAr 10200, with or without the addition of human complement (5% final concentration) (Cederlane) at 4 °C overnight. Monoclonal antibodies were from the USAMRIID repository. After the incubation, the antibody and virus mixtures were then added to 80−90% confluent SW13 cells in duplicate in six-well plates after the cell media

was removed (100 μl of sample was added per well), and the plates were incubated at 37 °C with 5% $CO_2$ for 1 h and rocked every 15 m to prevent drying of the monolayer. A 2 ml per well primary overlay was added with a final concentration of 1× phenol red free EMEM (Quality Biologics), 5% FBS (HyClone), and 1% Glutamax (Gibco), and 0.5% SeaKem ME agarose (Kemp) and the plates were incubated for 3 days at 37 °C with 5% $CO_2$. On the third day 2 ml of a 5% neutral red solution (SIGMA) in PBS was added to each well and incubated for 1–2 h before counting the plaques. A 50% reduction in plaques was calculated from the mAb-6D8 negative control.

### Surface presence of NP

A549 cells were infected with CCHFV strain IbAr 10200 for 48 h at an MOI of 1 in black 96-half-well optical plates (Corning). Following infection cells were fixed with methanol-free formaldehyde for one hour at room temperature. The cells were blocked with either solutions of 3% BSA (Sigma) and 0.1% Triton X-100 (Sigma) in PBS (Corning) to permeabilize, or 3% BSA in PBS without Triton x-100 to not permeabilize. The cells were then stained overnight at 4 °C, in either the permeabilizing or non-permeabilizing buffers described previously, with mAb-9D5 anti-CCHFV N, mAb-8A11 anti-Gc, or mAb-7D11 at a 1:200 dilution. These antibodies are from the USAMRIID monoclonal repository. The cells were then stained with Alexa 488 goat anti-mouse secondary Ab (Invitrogen; A-11001) at a dilution of 1:1000 in the same buffers as the primary antibody for one hour in the dark. After staining, all cells were fully submerged in 10% neutral buffered formalin (VAL Tech) for 24 h minimum, at +4 °C in the dark, to inactivate virus. Prior to imaging cells were additionally stained with Hoechst (Invitrogen) and Cell Mask Deep Red (Invitrogen; C10046) in PBS, at dilutions of 1:10,000 and 1:50,000, respectively. Two experiments were conducted, each with six replicates.

### Statistical analysis

Survival statistics utilized the log-rank test. Significance levels were set at a *P* value less than 0.05. All analyses were performed using GraphPad Prism 7 software.

## Data availability

All data generated or analyzed during this study are included in this published paper, source data file and supplementary files. Source data are provided with this paper. Data that support the findings of this study are also available from the corresponding author upon request. Mass spectrometry data was deposited in the MassIVE database database under accession code MSV000092981.

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

## Acknowledgements

We thank the Veterinary Medicine Division and Nicolas Di Paola for technical assistance. A.R.G., X.Z., C.R.C., M.D.W., K.M.R., S.P.O., L.H.C., C.J.F., and J.W.G. were supported by a Military Infectious Disease Research Program grant (MI240084). V.M., E.K., E.B., and S.D.P. were supported by NIAID (R01AI151006) and the Defense Threat Reduction Agency (HDTRA12210007). Opinions, interpretations, conclusions, and recommendations are those of the authors and not necessarily endorsed by the Centers for Disease Control and Prevention, the U.S. Army, or the Department of Defense.

## Author contributions

A.R.G., V.M., M.D.W., L.H.C., S.D.P., and J.W.G. designed research. A.R.G., V.M., X.Z., C.R.C., M.D.W., K.M.R., S.P.O., L.H.C., C.J.F., S.D.P., and J.W.G. performed research. A.R.G., V.M., X.Z., C.R.C., M.D.W., K.M.R.,

S.P.O., L.H.C., E.K., C.J.F., E.B., S.D.P., and J.W.G. analyzed the data. A.R.G. and J.W.G. wrote initial drafts of the manuscript. All authors contributed to the final manuscript.

## Competing interests

A.R.G. and J.W.G. have a provisional patent application regarding NP-targeting antibodies and their use to protect against CCHFV infection. The remaining authors declare no competing interests.
