## [Peer Review File · Nature Communications]

Reviewers' Comments:

Reviewer #1:

Remarks to the Author:

The manuscript by Garrison et al describes a series of experiments assessing the protective efficacy of an anti-NP based antibody against CCHFV infection in a commonly used immune deficient mice model. The experiments outlined are appropriate for the goals of the study however the findings and their direct application are over-estimated by the authors. The title states the nucleocapsid is "an important target of protective monoclonal antibodies" yet the authors only obtain 50% survival and only when the antibody is administered 24 hours before viral challenge. While these findings suggest the NP is a unique or perhaps unexpected target for protective antibodies (though this has been documented by several other labs for many other viruses), their findings argue against it being significant, at least on its own. As outlined below, the analysis of some of the data from these protection experiments may also require re-evaluation as it seems at least with respect to median time to death, the data is skewed and presents a more beneficial effect of treatment.

There is some discussion towards overcoming the limitation of this modality being only prophylactically effective through use in high exposure settings. While this discussion is valid, it needs to take into account recent studies on direct acting antivirals, particularly a recent article by Tipih et al, who have shown that ribavirin and favipiravir treatment is effective at preventing disease in mouse models even when treatment is initiated at the onset of disease signs.

Overall, this paper is interesting and the findings warrant publication; however the data presented do not offer any advancement towards medical countermeasures for the treatment of CCHFV disease. I recommend the authors soften the language towards this aspect which is not substantiated by their findings and re-submit.

Additional comments:

1. Results (lines 112-113): The authors state the median time to death of treated mice was 14 days but according to figure 1A, all the treated mice that succumbed to CCHFV infection were dead before day 10. It seems that the authors may have artificially prolonged the time to death of treated mice by including mice that were euthanized at the end of the experiment. If so, this would over-estimate the statistical significance of the perceived delay in time to death. Please review and correct.
2. Given that 50% of the mice in the first experiment were not protected by antibody treatment, the authors should explicitly state what the clinical score of the liver displayed for the treatment group at 4 and 8 dpi in Figure 1B.
3. Figure S2: The spleen data seems more informative for the protection afforded by the antibody treatment and should be included in a combined figure with liver H&E and ISH in the main text. Did these spleens come from the same animals as the liver shown in Figure 1B? What was the animals individual clinical scores?

Reviewer #2:

Remarks to the Author:

The manuscript titled "Crimean-Congo hemorrhagic fever virus nucleocapsid is an important target of protective monoclonal antibodies in mice" by Garrison and colleagues described an antibody, which targets the CCHFV NP protein, cross-protects two types of model mice from diverse CCHTV infections by limiting early viral spread to liver and spleen, and the protection is independent of Fc and complement functionalities. However, this antibody does not protect if used after viral exposure. The finding of NP antibodies capable of protect from infection is interesting even though the exact mechanisms are not known. I have only minor comments:

1. What is the background of antibody mAb-9D5? The authors may need feed us a little bit more details.
2. Since the ADCC and complement are not required for protection, how is this antibody able to prevent from infection? Preventing new viral particle from assembling before a stable infection is established?
3. The authors used mass spec to identify the potential binding epitope, it is intriguing to see that the identified region is highly conserved with positively charged residues and negatively charged residues on the two sides of the helix alpha8. With the power of AlphaFold2, could the authors model 9D5 structure and display the electro-static potential surface to see if there is charge complementarity on

the surface of paratope? If the authors can perform a negative stain EM if such facilities are available, that would confirm the potential binding mode.

4. Typo/errors in Figures:

- a. There is a name "mAb-13G8" in both Figure 2 and Figure 6, please correct.
- b. Figure 5, for Erve virus, there are "OR" for all columns, please explain what does that mean.
- c. Table S2: nM shift for 9D5 on Afg09, the SD showed +/-02, please correct the number.

Reviewer #3:

Remarks to the Author:

The paper by Garrison and colleagues provides compelling evidence of protection (in a murine model) conferred by an NP-specific monoclonal antibody against CCHFV challenge. Whilst protection wasn't absolute (at the concentrations used) the effect was statistically significant, and not conferred by direct virus neutralisation nor ADCC/complement effector mechanisms. The discussion raises possible mechanisms of protection, some considered unlikely by the authors (eg TRIM21), although the exact mechanism will require further research. The discussion also raises other unknowns, eg better understanding of the 9D5 epitope, but I agree that this is, and mechanism of action are beyond the current study.

1. Figure 1 panel B and Fig 4 – are the % weight change mean or median values, please state in legend. Also add SE/SD error bars. [Notwithstanding that a statistics section was included in the methods.]
2. Line 120 – Figure 1B does not relate to liver disease
3. Fig 2 – please state in legend whether mean or median and what the error bars relate to.
4. Fig 2 B – the isotype control mice seemed to have delayed weight loss and death-rate – this isn't mentioned nor possible explanation given.
5. Fig 5B – it might be informative to add the percentage amino acid identity across residue 184-208 between each strain and IbAr 1200 to panel B.
6. Fig S4 & Table S2 – I am not fully aware of BLI, but shouldn't a negative control mAb be added (eg Gc-specific) to reassure that wavelength shifts are due to 'real' binding events
7. With regards to statistical testing, were the tests parametric or non-parametric, and if the former, were normality tests performed?

REVIEWER COMMENTS

Reviewer #1 (Remarks to the Author):

The manuscript by Garrison et al describes a series of experiments assessing the protective efficacy of an anti-NP based antibody against CCHFV infection in a commonly used immune deficient mice model. The experiments outlined are appropriate for the goals of the study however the findings and their direct application are over-estimated by the authors. The title states the nucleocapsid is “an important target of protective monoclonal antibodies” yet the authors only obtain 50% survival and only when the antibody is administered 24 hours before viral challenge. While these findings suggest the NP is a unique or perhaps unexpected target for protective antibodies (though this has been documented by several other labs for many other viruses), their findings argue against it being significant, at least on its own. As outlined below, the analysis of some of the data from these protection experiments may also require re-evaluation as it seems at least with respect to median time to death, the data is skewed and presents a more beneficial effect of treatment.

There is some discussion towards overcoming the limitation of this modality being only prophylactically effective through use in high exposure settings. While this discussion is valid, it needs to take into account recent studies on direct acting antivirals, particularly a recent article by Tipih et al, who have shown that ribavirin and favipiravir treatment is effective at preventing disease in mouse models even when treatment is initiated at the onset of disease signs.

Overall, this paper is interesting and the findings warrant publication; however the data presented do not offer any advancement towards medical countermeasures for the treatment of CCHFV disease. I recommend the authors soften the language towards this aspect which is not substantiated by their findings and re-submit.

Thank you for your comments and we concur that these findings are “interesting and..warrant publication”. The ability of anti-NP 9D5 mAb to provide 80% protection when given as a prophylactic against a strain from a human lethal case shows this targeting strategy has great promise as a safer medical countermeasure compared to other treatments that have adverse impacts on liver functionality. No previously reported study definitively shows NP was a protective humoral target for CCHFV. In this regard, this manuscript is a trailhead, identifying a new antibody target and reporting its promise to the field. Furthermore, we identified potentially more potent antibodies in the supplemental figures which may show improved efficacy, and hopefully improved efficacy will translate to success with a therapeutic regimen. These initial findings, of protection afforded by targeting NP with an antibody, is a first for Nairoviruses and many viruses in the Bunyaviridae order. Previously, within this order only LCMV (an arenavirus)) showed protection with anti-NP antibodies, and in the manuscript we state that NP targeting antibodies have been shown to protect against several other virus groups (lines 274-276). As described in the discussion, the inclusion of an anti-NP antibody may provide a compliment to any antibody medical countermeasure combination, as treatments in humans would more likely be a cocktail and not a single antibody. In addition, stabilized long acting antibodies have merit as passive vaccination in providing protection in high risk populations with low adverse events, for example protection against COVID-19 disease with Evushield (AZD7442). The statistical explanation below addresses the question of experimental significance, and the increased protection (80 %) against a human relevant strain of CCHFV

shown in manuscript (Figure 6) address the prophylactic significance above 50% question. Our interpretation of medical countermeasures includes active and passive vaccines and other prophylactic treatments as well as therapeutic treatments, so we have made this more clear in the text and title.

In regard to ribavirin specifically, the effectiveness of ribavirin in mice is variable given the strain and model, it is protective with a mouse adapted virus in wild type male mice (Tipih et al referenced above by the reviewer); however, it was not protective against a non-adapted CCHFV in IFNAR^{-/-} mice (Hawman et al 2018). Ribavirin treatment has mixed efficacy in humans as well and would not be used prophylactically (Ceylan B, 2013 doi:10.1016/j.ijid.2013.02.030; Dokuzoguz, B et al 2013 doi:10.1093/cid/cit527; Ergonul, et al; 2004, doi: 10.1086/422000; Koksai, et al 2010, doi.org/10.1016/j.jcv.2009.11.007)

Several things were included in the text were added to address these points:

1: Modified the title tone it down; "Crimean-Congo hemorrhagic fever virus nucleocapsid protein is a target of protective monoclonal antibodies in mice."

2. Line 59: added "when given prophylactically" to the Significance statement.

Additional comments:

1. Results (lines 112-113): The authors state the median time to death of treated mice was 14 days but according to figure 1A, all the treated mice that succumbed to CCHFV infection were dead before day 10. It seems that the authors may have artificially prolonged the time to death of treated mice by including mice that were euthanized at the end of the experiment. If so, this would over-estimate the statistical significance of the perceived delay in time to death. Please review and correct.

The typical time frame for our CCHFV mouse studies is 10-15 days, with mean time to death in untreated animals occurring around day 5 (Lindquist, et al e01083-18 (2018)., Golden et al Sci. Adv. 2019; 5: eaaw9535; Golden et al Plos Path; Pathog 18(5): e1010485.; Bente, et al J Virol. 2010 Nov;84(21). CCHFV mouse studies in our lab and others are routinely carried out to ensure that all treated animals have fully recovered as measure by weight gain at or above starting weight, and we utilized the same study design here. This design will ensure we catch all animals that may succumb to disease if a treatment only delayed severe disease. Our delay to ending the study in the first experiment was purposeful and provided due diligence that the mice treated with MAb-9D5 recovered. Note that our experimental design (under the guidance of our requisite animal studies statistical group) utilizes significant numbers to determine significant survival with a minimum of 50% with 0% in the control group at the end of the observation period, and does not take into account time of death. Using a one-tailed Fisher's exact test to compare survival rates between the control and treatment groups, this experiment requires a sample size of 10 animals per group for adequate (>78%) power. This sample size would allow the experimenter to detect a minimum efficacy rate of 50% (5/10 surviving) in the treatment group compared to 0% (0/10 surviving) in the control group at a 95% confidence level. This test would have a p value of 0.0163 in the first experiment. The log-rank test used in each experiment is the

standard test and weights all observations the same. The median survival time does not translate to “time-to-death” exactly, we adjusted the language in the manuscript to make this more clear (lines 136-138): “ In addition to the increase in percent surviving in the mAb-9D5 treated group (50%) versus the control group, there was also a delay in death in the treated group, with a median survival of 14 days compared to 5 days in the isotype control group.”

2. Given that 50% of the mice in the first experiment were not protected by antibody treatment, the authors should explicitly state what the clinical score of the liver displayed for the treatment group at 4 and 8 dpi in Figure 1B.

All treated mice scored 0 on day 4, 1 isotype control mouse out of 10 scored 1 (slight ruffled appearance) on day 4. On day 8 dpi, 1 mouse scored a 1 (slight ruffled appearance) and the remaining mice scored 0. Furthermore, we show the pathology in the mice, with particular emphasis on the liver (Fig 1B) and the pathology shown is representative of all of the samples taken at that time-point, as stated in the figure legend. We think this is more complete than clinical scores as it provides clear evidence that the antibody delays disease, but does not fully prevent it. Clinical scores for CCHFV in mice is well known to be difficult, as mice rapidly succumb to disease within hours. Thus the pathology, including infiltrating inflammatory cells provides a clearer picture.

3. Figure S2: The spleen data seems more informative for the protection afforded by the antibody treatment and should be included in a combined figure with liver H&E and ISH in the main text. Did these spleens come from the same animals as the liver shown in Figure 1B? What was the animals individual clinical scores?

We did not ear tag the mice for individual scores, the liver/spleen shown was representative of all 4 animals per group taken per timepoint within a group. As stated above, only one animal had a clinical score of 1 (slightly ruffled appearance) the remaining scored 0 on day 4 and day 8.. Other than weight loss, which is a separate measurement from clinical scoring, clinical scoring is typically minimal and mice that appear normal often succumb after a few hours , most likely due to rapid onset of multi-organ failure due to the proinflammatory response (Lindquist, et al e01083-18 (2018)). This clinical score information has been added to lines 153-155. Because the liver is the primary target and incurs substantial injury from CCHFV in mice and humans, it is the most important component to examine by pathology. The spleen, while useful, is not a vital organ.

Reviewer #2 (Remarks to the Author):

The manuscript titled “Crimean-Congo hemorrhagic fever virus nucleocapsid is an important target of protective monoclonal antibodies in mice” by Garrison and colleagues described an antibody, which targets the CCHFV NP protein, cross-protects two types of model mice from diverse CCHTV infections by limiting early viral spread to liver and spleen, and the protection is independent of Fc and complement functionalities. However, this antibody does not protect if used after viral exposure. The finding of NP antibodies capable of protect from infection is interesting even though the exact mechanisms are not known. I have only minor comments:

1. What is the background of antibody mAb-9D5? The authors may need feed us a little bit more details.

The development of the mouse mAb-9D5, and the reference with additional details of the development of the anti-CCHFV mAb USAMRIID collection, was added to the materials and methods section. The antibody isotype was also added. All of the antibodies used were murine antibodies.

2. Since the ADCC and complement are not required for protection, how is this antibody able to prevent from infection? Preventing new viral particle from assembling before a stable infection is established?

As the reviewer noted and despite our initial efforts, the exact mechanism of action studies are beyond the current scope of work, and we are actively studying the potential mechanism. Without additional experimentation any response would be pure speculation, however, we do state the TRIM21 could be a suspect given its role in old world arenavirus protection.

3. The authors used mass spec to identify the potential binding epitope, it is intriguing to see that the identified region is highly conserved with positively charged residues and negatively charged residues on the two sides of the helix alpha8. With the power of AlphaFold2, could the authors model 9D5 structure and display the electro-static potential surface to see if there is charge complementarity on the surface of paratope? If the authors can perform a negative stain EM if such facilities are available, that would confirm the potential binding mode.

Thank you for your comments and pertinent suggestions. Originally, we had the same thought and did try to pursue that angle. Regrettably, out of the 7 X-ray structures of NP originating from CCHFV (4), Hazara (2), and Kupe (1), the 10 residues in question from ~184-194 are not structurally resolved. The Erve NP is the only one with this region resolved and does not interact with 9D5. Hence, there is currently no working complete structures of NP homologues that have been shown to bind with 9D5. Not surprisingly, AlphaFold2, which is based on ML from existing structures, produced models of this region that had AlphaFold per-residue model confidence score (pLDDT) lower than 50 for the previously unresolved area. As per the AlphaFold website (<https://alphafold.ebi.ac.uk/faq>), "regions with pLDDT < 50 often have a ribbon-like appearance and should not be interpreted."

Naturally, for modeling a FAB via AlphaFold2 fared better, but the model didn't reveal any distinct charged patches that would indicate a binding mode. This is also not too surprising as most CDR loops in mAbs tend to undergo changes when binding. The statistics showing AlphaFold2 models being meritless for the NP region in question and no clear 3D binding mode between the NP-mAb with existing data, we viewed that it would be highly speculative to include these modelling figures.

The MS method utilized to define the binding epitope of mAb-9D5 is a tried-and-true method

1. Pimenova T, Nazabal A, Roschitzki B, Seebacher J, Rinner O, Zenobi R. Epitope mapping on bovine prion protein using chemical cross-linking and mass spectrometry. *J Mass Spectrom.* 2008 Feb;43(2):185-95. doi: 10.1002/jms.1280. PMID: 17924399.
2. Graziadei A, Rappsilber J. Leveraging crosslinking mass spectrometry in structural and cell biology. *Structure.* 2022 Jan 6;30(1):37-54. doi: 10.1016/j.str.2021.11.007. Epub 2021 Dec 10. PMID: 34895473.

3. Piersimoni L, Kastritis PL, Arlt C, Sinz A. Cross-Linking Mass Spectrometry for Investigating Protein Conformations and Protein-Protein Interactions—A Method for All Seasons. *Chem Rev.* 2022 Apr 27;122(8):7500-7531. doi: 10.1021/acs.chemrev.1c00786. Epub 2021 Nov 19. PMID: 34797068.
4. Petrotchenko EV, Nascimento EM, Witt JM, Borchers CH. Determination of Protein Monoclonal-Antibody Epitopes by a Combination of Structural Proteomics Methods. *J Proteome Res.* 2023 Sep 1;22(9):3096-3102. doi: 10.1021/acs.jproteome.3c00159. Epub 2023 Aug 1. PMID: 37526474; PMCID: PMC10476242.

this information references were added to the manuscript to reflect this (line 228-229).

4. Typo/errors in Figures:

- a. There is a name “mAb-13G8” in both Figure 2 and Figure 6, please correct.

Corrected.

- b. Figure 5, for Erve virus, there are “OR” for all columns, please explain what does that mean.

This has been changed to NB for not binding to keep it consistent with other figures, and the definition was added to the figure legend.

- c. Table S2: nM shift for 9D5 on Afg09, the SD showed +/-02, please correct the number.

The correct value is 0.02. This was fixed in the Table, which is now Table S3.

Reviewer #3 (Remarks to the Author):

The paper by Garrison and colleagues provides compelling evidence of protection (in a murine model) conferred by an NP-specific monoclonal antibody against CCHFV challenge. Whilst protection wasn't absolute (at the concentrations used) the effect was statistically significant, and not conferred by direct virus neutralisation nor ADCC/complement effector mechanisms. The discussion raises possible mechanisms of protection, some considered unlikely by the authors (eg TRIM21), although the exact mechanism will require further research. The discussion also raises other unknowns, eg better understanding of the 9D5 epitope, but I agree that this is, and mechanism of action are beyond the current study.

1. Figure 1 panel B and Fig 4 – are the% weight change mean or median values, please state in legend. Also add SE/SD error bars. [Notwithstanding that a statistics section was included in the methods.]

Thank you for catching our error. These are group weights, not individual weights. This information was added to all the legends that include weight data. The statistical mention of weight comparisons was incorrectly added and has been removed.

2. Line 120 – Figure 1B does not relate to liver disease

Figure 1B shows H&E and ISH staining of the infected livers. This histological analysis directly examines liver injury and was assessed by a board-certified veterinary pathologist.

3. Fig 2 – please state in legend whether mean or median and what the error bars relate to.

The text in the legend was updated to state: The bars are the mean of three replicates and the error bars are the standard deviations of the replicates.

4. Fig 2 B – the isotype control mice seemed to have delayed weight loss and death-rate – this isn't mentioned nor possible explanation given.

The same isotype control was used in the experiment in Figure 1 and Figure 2. There is no significant difference in the survival between the isotype controls between the two experiments, and there will be some minor variability between experiments. The median survival in the isotype is 5 dpi in Figure 1 and 4.75 dpi in Figure 2. On day 4 of both experiments the isotype weight loss is -7.63% in Figure 1 and -5.5% in Figure 2, and group weights were performed so statistics are not performed on weights, the weight difference is minimal. We included this isotype control in each experiment to account for any variability between experiments. The 50% survival in the mAb 9D5 was consistent across both experiments utilizing IFNAR^{-/-} mice and IbAr 10200.

5. Fig 5B – it might be informative to add the percentage amino acid identity across residue 184-208 between each strain and IbAr 1200 to panel B.

A supplemental table was added with this information, now Table S2.

6. Fig S4 & Table S2 – I am not fully aware of BLI, but shouldn't a negative control mAb be added (eg Gc-specific) to reassure that wavelength shifts are due to 'real' binding events

Because anti-NP CCHFV antibodies did not bind Erve virus, this functions as our negative control. This is ideal, as it is an authentic nairovirus NP. Gc would have been less ideal as it is a glycoprotein with protein chemistry much different than the nucleoprotein. Note the supplemental data is now in Table S3.

7. With regards to statistical testing, were the tests parametric or non-parametric, and if the former, were normality tests performed?

The log-rank test for survival used is non-parametric, which is a distribution-free test and does not require a normal distribution. We used group weights, so we removed the reference to weight statistics in the materials and methods.

REVIEWERS' COMMENTS

Reviewer #2 (Remarks to the Author):

The authors have adequately addressed my comment. thanks!

Reviewer #3 (Remarks to the Author):

None